# The Efficiency of Sclerotherapy for the Management of Endometrioma: A Systematic Review and Meta-Analysis of Clinical and Fertility Outcomes

**DOI:** 10.3390/medicina59091643

**Published:** 2023-09-11

**Authors:** Carlo Ronsini, Irene Iavarone, Eleonora Braca, Maria Giovanna Vastarella, Pasquale De Franciscis, Marco Torella

**Affiliations:** Department of Woman, Child and General and Specialized Surgery, University of Campania “Luigi Vanvitelli”, 80138 Naples, Italy; carlo.ronsini@unicampania.it (C.R.); ireneiavarone2@gmail.com (I.I.); eleonorabraca9@gmail.com (E.B.); mariagiovanna.vastarella@studenti.unicampania.it (M.G.V.); pasquale.defranciscis@unicampania.it (P.D.F.)

**Keywords:** endometriosis, ovary, sclerotherapy, ethanol, pregnancy

## Abstract

*Background and Objectives*: The most common sites of implantation of endometriotic tissue are the ovaries. Endometriomas are present in most cases of endometriosis (up to 45%). Although laparoscopic cystectomy is the standard of care in endometrioma, new strategies have been set up to minimize iatrogenic injuries to ovarian tissue. Sclerotherapy consists of injecting alcohol into the endometrioma to denature the amino acidic components of its pseudocapsule. The aim of this systematic review and meta-analysis is to compare clinical and pregnancy outcomes in surgery and sclerotherapy. *Materials and Methods*: Following the recommendations in the Preferred Reporting Items for Systematic Reviews and Meta-Analyses (PRISMA) statement, we systematically searched PubMed, EMBASE, Scopus, Google Scholar, Clinical-trials.gov, and the Cochrane Central Register of Controlled Trials databases in January 2023, adopting the string “Endometriosis and sclerotherapy”. We made no limitations on the country and year of publication. We included the studies containing Success Rate (SR), Recurrence Rate (RR), Pregnancy Rate (PR) before and after the procedure. We used comparative studies for meta-analysis. *Results*: A total of 29 studies fulfilled inclusion criteria, 7 retrospective observational studies and 22 prospective studies. Eight comparative studies were enrolled in meta-analysis. Patients were analyzed concerning the number of recurrences and pregnancies in surgery, and compared with sclerotherapy. Four studies showed SR > 80.0%, and only two had SR < 80.0%, of which one consisted of tetracycline instillation. Only 1 study had 100% PR, the other 14 reported PR > 30.0%, whereas six had PR < 30.0%, of which one showed 0.0% PR with ethanol injection at two-thirds of the cyst fluid volume. Meta-analysis highlighted a non-significant lower incidence of recurrence in the surgery group compared to the sclerotherapy group (*p* = 0.87). In parallel, the surgery group showed a non-significant better PR than the sclerotherapy group (*p* = 0.08). *Conclusions*: Despite sclerotherapy having a minor incidence of postoperative complications compared to surgery, the latter is associated with a lower RR and better PR. However, those data assert the importance of a targeted therapy according to preoperative conditions and reproductive potential.

## 1. Introduction

Endometriosis is an estrogen-dependent gynecological disorder affecting 10% of childbearing-age women [1]. It consists of the migration of endometrial tissue (stroma plus glands) outside the uterine cavity. The most common sites of implantation are the ovaries, which could result in a superficial or deep variant [1]. In particular, the deep form of ovarian endometriosis is represented by endometrioma—an ovarian cyst encapsulated by endometrial tissue and containing dark brown fluid, deriving from blood catabolism [2]. Endometriomas are present in almost 45% of cases of endometriosis [2]. Nowadays, the standard of care in endometrioma is laparoscopic cystectomy [3]. Given the chronic nature of endometriosis, the treatment strategy could be repeated; however, surgery damages residual ovarian parenchyma. The damage may result in a reduction in the patient’s fertile potential, with a risk of infertility and early menopause. For this reason, iatrogenic damage should be minimized, and a strategy should be devised to resolve the clinical episode with the least possible sequelae. The surgical instruments and coagulation for hemostasis could damage the ovarian tissue [4,5,6]. So, various research groups have studied alternative strategies for treating endometrioma. One of the most popular is the use of sclerotherapy. The rationale for sclerotherapy is to destroy the pseudocapsule of the endometrioma by instilling alcoholic substances inside it [7]. Sclerotherapy has been considered a cost-effective and safe option to preserve healthy ovarian tissue with a low incidence of Clavien-Dindo complications [8,9,10,11,12]. The practitioner should perform an injection of a sclerosing substance into the cyst, to proceed with washing or retention [13]. Hydrosoluble dehydrating antiseptic fluids are able to denature the proteins upon the envelop of microorganisms and dissolve their capsule’s lipids. In most cases, the ethanol is preferred to sclerosing substances because it has shown better outcomes in the management of renal and hepatic cysts [7]. Sclerotherapy can be practiced through the use of both general or local anesthetics, and it is well-tolerated by the patients. The feasibility in programming, cost-effectiveness, and rapid recovery after the technique suggest that it can be considered the standard of care for minimally invasive interventions, like ethanol sclerotherapy of ovarian endometrioma [7]. Nevertheless, to date, no randomized clinical trials have proven the greater effectiveness of one technique over the other, either in terms of reducing the recurrence rate or improving pregnancy rates. The aim of the present systematic review and meta-analysis was to evaluate clinical and pregnancy outcomes of sclerotherapy in managing ovarian endometriomas, compared to standard cystectomy.

## 2. Materials and Methods

The present study was exempted from ethical approval, because it does not include interventions on human subjects. The methods for this study were specified a priori based on the recommendations in the Preferred Reporting Items for Systematic Reviews and Meta-Analyses (PRISMA) statement [14]. The review is registered on PROSPERO as ID394764.

### 2.1. Search Method

We performed systematic research for records about the use of sclerotherapy in managing ovarian endometriomas in PubMed, EMBASE, Scopus, Google Scholar, Clinical-trials.gov, and the Cochrane Central Register of Controlled Trials in January 2023. We made no restriction on country, nor year of publication, and considered only entirely English published studies. We adopted the following string of idioms in each database to identify studies fitting to our review’s topic: “Endometriosis and sclerotherapy”.

### 2.2. Study Selection

Study selection was made independently by I.I. and M.G.V. In case of discrepancy, C.R. decided on inclusion or exclusion. Inclusion criteria were: (1) studies that included patients with at least one ovarian endometrioma, treated with sclerotherapy and/or surgery; (2) studies reporting at least one outcome of interest: Success Rate (SR), Recurrence Rate (RR); Pregnancy Rate (PR) after the procedure (3) peer-reviewed articles, published originally. We excluded: non-original studies, pre-clinical trials, animal trials, abstract-only publications, and articles in a language other than English. If possible, the authors of studies that were published as conference abstracts were tried to be contacted via e-mail and asked to provide their data. We mentioned the studies selected and all reasons for exclusion in the Preferred Reporting Items for Systematic Reviews and Meta-Analyses (PRISMA) flowchart (Figure 1). We assessed all included studies concerning potential conflicts of interest.

### 2.3. Statistical Analysis

Heterogeneity among comparative studies was tested using the Chi-square test and I-square tests [15]. Risk rates and 95% confidence intervals (CI) were used for dichotomous variables. Fixed-effect models conducted statistical analysis without significant heterogeneity (I2 < 50%), or models if I2 > 50%. SR, RR, complications rate, and PR were the clinical outcomes. SR (number of healed patients among the number of attempts with sclerotherapy or surgery), RR (the number of relapses during follow-up), and PR (number of pregnant patients over the total of women who tried to conceive) were calculated in each study as percentages. The chi-square test was used to compare continuous variables.

### 2.4. Quality Assessment

We assessed the quality of included studies using the Newcastle-Ottawa Scale (NOS) [16]. That assessment scale uses three broad factors (selection, comparability, exposure), with scores ranging from 0 (lowest quality) to 9 (best quality). Two Authors (C.R. and I.I.) independently rated the studies’ quality. Any disagreement was subsequently resolved by discussion or consultation with P.F.G. We reported NOS Scale in Appendix A. We used a funnel plot analysis to establish publication bias. We employed Egger’s regression test to determine the asymmetry of the funnel plots (Appendix B and Appendix C).

## 3. Results

### 3.1. Studies’ Characteristics

After the database search, 134 articles matched the searching criteria. After removing records with no full-text, duplicates, and wrong study designs (e.g., reviews), 37 were suitable for eligibility. Of those, 29 matched inclusion criteria and were included in the systematic review. A total of 21 of them were non-comparative, single-armed studies evaluating only sclerotherapy [17,18,19,20,21,22,23,24,25,26,27,28,29,30,31,32,33,34,35,36,37]. The other eight studies were comparative studies between surgery and sclerotherapy and were included in quantitative analysis (Figure 1) [37,38,39,40,41,42,43,44,45]. The countries where the studies were conducted, the publication year range, the studies’ design, number of participants, substance instilled, and procedure characteristics are summarized in Table 1. The quality of all studies was assessed by NOS [16] (Appendix A). Overall, the publication years ranged from 1997 to 2022 [17,18,19,20,21,22,23,24,25,26,27,28,29,30,31,32,33,34,35,36,37,38,39,40,41,42,43,44,45]. In total, 1642 patients with endometriomas were included [17,18,19,20,21,22,23,24,25,26,27,28,29,30,31,32,33,34,35,36,37,38,39,40,41,42,43,44,45]. The follow-up (FU) period ranged from 1.5 to 84 months on average [17,18,19,20,21,22,23,24,25,26,27,28,29,30,31,32,33,34,35,36,37,38,39,40,41,42,43,44,45].

### 3.2. Outcomes

A total of 1642 patients were included in the review [17,18,19,20,21,22,23,24,25,26,27,28,29,30,31,32,33,34,35,36,37,38,39,40,41,42,43,44,45]. Not all of the 29 selected studies presented data about clinical outcomes—SR, RR—and pregnancy outcomes—PR before and after the procedure. It consisted of ethanol, tetracycline or methotrexate instillation at specific dilutions. Instilled volume ranged from 20% to 80% of the cyst fluid volume for 5–20 min or left in situ. Except for 12, other 17 studies presented SR data in sclerotherapy [18,19,20,21,22,24,26,27,29,30,31,33,35,36,39,41]. A total of 11 studies reported a 100% SR [19,21,22,24,29,30,31,35,36,39,43], 4 studies showed an SR > 80.0% [18,26,27,33], whereas only two studies had an SR < 80.0% [20,30], of which one consisted of tetracycline instillation [19]. Moreover, except for 5, the other 24 studies reported RR data in sclerotherapy [17,18,19,20,21,22,23,24,26,27,28,29,30,31,32,33,34,35,37,39,41,42,43]. A total of five studies reported a 0.0% RR at 12 months FU [20,27,30,31,43], and 15 studies showed a RR between 0.0% and 30.0% [19,21,22,23,24,26,28,29,30,33,34,35,37,39,41]. In contrast, only four studies had a RR > 30% [17,18,31,43], of which one consisted of 1% tetracycline instillation at 20% of the cyst fluid volume and left in situ, and one in 30 mg methotrexate instillation diluted in 3 mL saline solution [17,18]. Secondarily, pregnancy outcomes were also evaluated: in 21 studies, it was feasible to extract data about PR in sclerotherapy [17,19,20,22,23,24,25,28,29,31,32,33,34,37,38,39,40,41,42,44]. Only 1 study had a 100% PR [31], the other 14 studies reported a PR > 30.0% [17,19,20,22,28,30,32,33,37,38,39,40,42,44], whereas six studies had a PR < 30.0% [23,24,25,29,34,41], of which one showed a 0.0% PR with ethanol injection at two-thirds of the cyst fluid volume [29]. Those results are summarized in Table 2 and Table 3.

### 3.3. Meta-Analysis

The eight studies comparing surgery and sclerotherapy were enrolled in the meta-analysis, exploring outcomes about recurrence and pregnancy. Four studies reported data about recurrences events. A total of 303 patients were analyzed, 148 in the surgery arm and 155 in the sclerotherapy arm. A total of 17 recurrences occurred in the surgery group vs. 27 recurrences in the sclerotherapy group. Because of the high heterogeneity (I262%; *p* = 0.05), a random-effects model was applied.

The surgery group showed a comparable incidence of recurrence than the sclerotherapy group (OR 0.87 [95% CI 0.18–4.32] *p* = 0.87) (Figure 2).

We performed a second analysis of the pregnancy rate. Seven of the eight comparative studies reported useful data. There was a total of 204 patients for the surgical group and 166 for the sclerotherapy group. Only patients attempting to become pregnant were considered in the second analysis. A total of 370 patients were analyzed, 204 in the surgery arm and 166 in the sclerotherapy arm. A total of 67 pregnancies occurred in the surgery group, vs. 72 recurrences in the sclerotherapy group. Because of the high heterogeneity (I260%; *p* = 0.02), a random-effects model was applied.

In addition, in that analysis, the surgical group documented a comparable pregnancy rate than the sclerotherapy (OR 0.47 [95% CI 0.21–1.09] *p* = 0.08) (Figure 3).

## 4. Discussion

Sclerotherapy is a method that is finding increasing acceptance in treating endometriomas. The reported data show that the SR is high when this technique is applied. In our opinion, the real comparison between the surgical technique and sclerotherapy should be made on the one hand on the RR, to optimize the chronification of the pathology and minimize treatment event, and, on the other hand, on the PR, to have an indirect view of how the treatment may have iatrogenic damage. The data reported in this systematic review show that sclerotherapy in endometriomas has a varied RR, ranging from 0.0% to 61.9%. However, except for a few studies, most literature data report recurrence rates <30%, over an average FU period of 1 year. However, the meta-analysis reported, even without statistical significance, a minor incidence of recurrence in the surgery arm compared to the sclerotherapy group (Figure 2). In parallel, despite the fact that PR reaches up to 100% in the sclerotherapy group, the surgery arm revealed better PR. Differences in scientific evidence in terms of RR and PR may be due to the methodology employed—in particular, sclerosing substance, but also concentration, instilled proportions, and duration of retention. The sclerosing agent seems to be relevant since sclerotherapy with tetracyclines or methotrexate had higher RR and lower PR. Tetracyclines and methotrexate act in the cell-cycle functions through molecular pathways, whereas ethanol injuries endometrioma’s pseudocapsule with a mechanism including cytotoxicity, thrombosis, and cells hypertonic dehydration [21,46,47]. Apparently, long ethanol retention times and small dilutions may be associated with the complete degradation of the pseudocapsule’s cytoarchitecture. However, due to the heterogeneity of data among studies, it was impossible to assess the ideal ethanol retention times or dilution percentages. Only one study, Noma et al. in 2001, reported a difference in RR deeply in favor of sclerotherapy beside the laparoscopic cystectomy (up to 97.6% vs. 67.0%—*p* < 0.05); however, the sample size was exiguous, and the study design was retrospective, including the years in which laparoscopic treatment was growing up [38]. In Appendix B, it is demonstrated, however, that results are homogeneous in the scientific literature, regarding the outcomes of recurrence. Regarding pregnancy outcomes, it must be considered that women with endometriomas have themselves lower ovarian reserve, perhaps due to a decreased response in Assisted-Reproductive Technology (ART) techniques [48]. However, surgical interventions themselves impact ovarian tissue and antral follicle count, revealed by modifications of Anti-Müllerian Hormone (AMH) [49,50,51,52]. In addition, repeated surgeries can also lead to adhesion syndrome, resulting in worsened fertility. Ethyl alcohol instillation can also damage follicles; however, the significant heterogeneity of studies makes it complicated to identify a dose and administration time that maximizes the result while minimizing the ovariotoxic effect. Our data analysis highlighted that there is homogeneity between the considered studies. It is shown through the Pregnancy Funnel Plot in Appendix C. It would be appropriate to understand whether adjuvant treatment options after sclerotherapy could improve PR and ameliorate ART response rates—for example, an increased dose of gonadotropins in women with preserved ovarian reserve. A further point to be made is about patient selection. In fact, sclerotherapy might be an alternative in cases of single endometriomas. In contrast, the endometrioma itself might be multiple or associated with pictures of severe endometriosis, where surgical approaches remain essential [53].

The main limitation of our systematic review and meta-analysis is the small number of randomized trials comparing sclerotherapy to surgery. Moreover, some articles included a poor number of patients, and heterogeneity among procedures could increase risks of bias and confounding. In addition, the term sclerotherapy embraces a range of methods that differ in substance and mode of administration, which is why our data are difficult to put into context. In any case, our results show a novelty in the landscape of current knowledge assessing lower RR and better PR in sclerotherapy compared to surgery [13]. Moreover, we highlight that sclerotherapy has a minimal incidence of Clavien-Dindo complications, and neither study showed Clavien-Dindo 3 or more sequelae after ethanol injection. Although, the overall complication amount in laparoscopy is low, and it is only constituted by intraoperative complexity due to previous surgeries [54].

Otherwise, since the presence of ovarian endometriomas is often associated with deep endometriosis, symptoms like chronic pelvic pain may be related mostly to deep adhesions [55]. Hence, amelioration of symptoms in women with ovarian endometrioma even after sclerotherapy could be improper. This could be considered as a further point of weakness of our study, and it would be feasible to investigate upon the detection of markers of deep infiltrating endometriosis [56,57]. Due to the comparable clinical and pregnancy outcomes of laparoscopic cystectomy and sclerotherapy in patients with endometriomas, the latter could be considered the standard of care in women with no evidence of deep endometriosis, given the cost-effectiveness and patient comfort of sclerotherapy. Unfortunately, there are poor data about the use of alcohol injection in young women with the desire of pregnancy. Although, according to recent evidence, ethanol injection shows better reproductive outcomes in women undergoing Assisted Reproductive technology [34,45,58,59]. Sclerotherapy may be a factual option also when a deeper surgery is contraindicated or in women with ovarian tissue injury or at risk of iatrogenic damage. Further perspectives may focus on the use of liquid biopsy or microbial composition to identify the best candidates for sclerotherapy [60,61]. Regarding the former, the detection of micro-RNAs (miRNAs)—a class of small RNA molecules, composed of 15–22 nucleotides each—may be associated with particular features of the endometrioma [60]. Regarding the latter, intestinal or Female Reproductive Tract (FRT) microbiota may be linked to fertility outcomes [61]. Those findings would be helpful in tailoring the management of patients with ovarian endometriomas, using molecular strategies in order to identify key characteristics to plan the best treatment modality.

## 5. Conclusions

Although sclerotherapy shows a low rate of complications in the postoperative period, surgery seems to be associated with decreased RR and higher PR. Our analysis demonstrates comparable incidence of recurrence and pregnancy rate between surgery and sclerotherapy, with homogeneity between the findings (Appendix B and Appendix C). Those data highlight the importance of a targeted therapy according to preoperative conditions and reproductive potential based on ovarian reserve in endometriosis-affected patients. Otherwise, sclerotherapy could be a safe and valid alternative in patients contraindicated for more complex surgery. Further evidence is needed to optimize and standardize the amount and duration of ethanol injection.

## Figures and Tables

**Figure 1 medicina-59-01643-f001:**
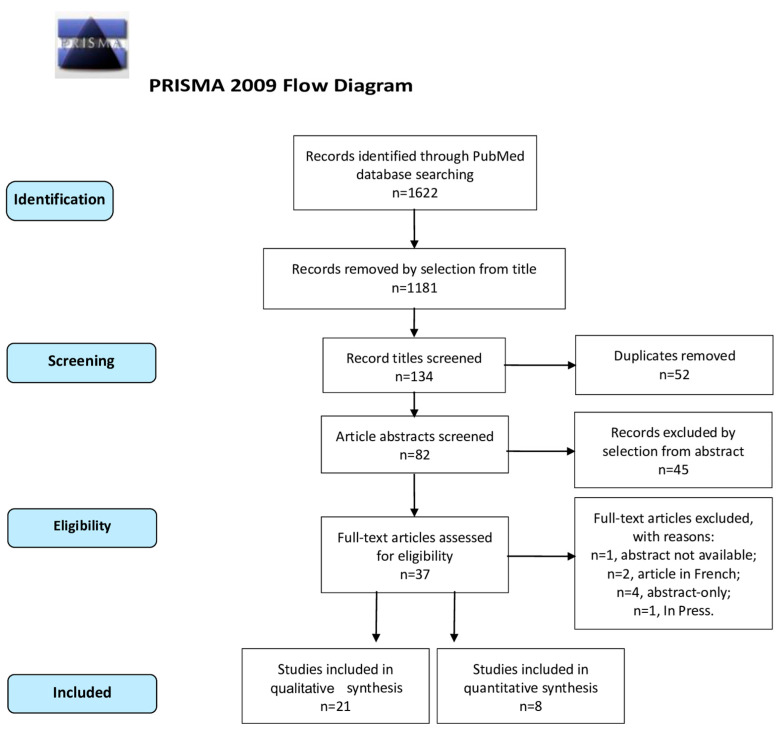
PRISMA (Preferred Reporting Items for Systematic Reviews and Meta-Analyses) Flow-Diagram.

**Figure 2 medicina-59-01643-f002:**
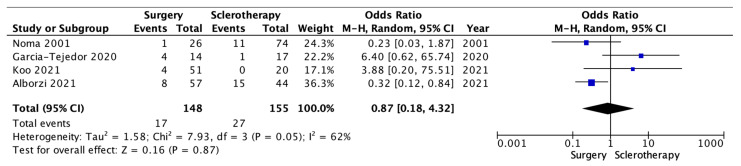
Recurrence Forest Plot [38,39,40,41,42,43,44,45].

**Figure 3 medicina-59-01643-f003:**
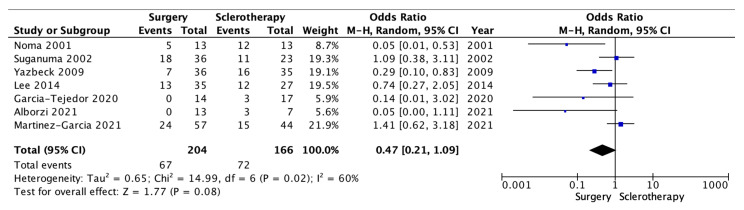
Pregnancy Forest Plot (*authors revised*).

**Table 1 medicina-59-01643-t001:** Characteristics of studies included.

* Single-Arm Studies *						
Author, Year of Publication	Country	Study Type	Period of Enrollment	No. of Patients	Procedure	Substance
Chang et al., 1997 [17]	China	Prospective cohort	1994–1996	32	† US-guided sclerotherapy	Tetracycline 1%. Instilled 20% of cyst fluid volume and left in situ.
Mesogitis et al., 2000 [18]	Greece	Prospective cohort	‡ N/A	11	US-guided sclerotherapy	Methotrexate 30 mg diluted in 3 mL saline solution
Koike et al., 2002 [19]	Japan	Retrospective cohort	1996–1998	110	US-guided sclerotherapy	Ethanol 50%. Instilled 100% of cyst fluid volume for 5 min.
Fisch et al., 2004 [20]	USA	Prospective cohort	N/A	11	US-guided sclerotherapy	Tetracycline 5%. Instilled volume 5–10 mL.
Agostini et al., 2006 [21]	France	Prospective cohort	2002–2003	14	Methotrexate sclerotherapy	Methotrexate 30 mg diluted in 3 mL saline solution
Ikuta et al., 2006 [22]	Japan	Prospective cohort	1999–2005	18	US-guided sclerotherapy	Ethanol 100% for 5 min.
Hsieh et al., 2008 [23]	Taiwan	Prospective cohort	2002–2005	108	US-guided sclerotherapy for 10 min (n = 78); retention sclerotherapy (n = 30)	Ethanol 95%. Instilled 80% of cyst fluid volume and ≤100 mL.
Gatta et al., 2010 [24]	Italy	Prospective cohort	2006–2008	50	US- or § LPS-guided sclerotherapy	Ethanol 95%. Instilled 100% of cyst fluid volume and left in situ.
André et al., 2011 [25]	Brazil	Prospective cohort pilot	2007–2010	21	US-guided sclerotherapy	Ethanol. Instilled 80% of cyst fluid volume and <60 mL for 5 min.
Shawki et al., 2011 [26]	Egypt	Randomized controlled trial	2007–2010	93	Methotrexate sclerotherapy (n = 93); aspiration (n = 95)	Methotrexate. 30 mg diluted in 3 mL normal saline solution.
Wang et al., 2011 [27]	China	Prospective cross-sectional	2006–2008	132	US-guided sclerotherapy for 10 min (n = 66); retention sclerotherapy (n = 66)	Ethanol 95%. Instilled 50% of cyst fluid volume.
Aflatoonian et al., 2013 [28]	Iran	Randomized controlled trial	2011–2012	20	US-guided sclerotherapy	Ethanol 98%. Instilled 80% of cyst fluid volume for 10 min.
Garcia-Tejedor et al., 2015 [29]	Spain	Prospective cohort	2016–2018	25	US-guided sclerotherapy	Ethanol. Instilled 66% of cyst fluid volume and <100 mL for 15 min.
Begum et al., 2015 [30]	Bangladesh	Prospective cohort	2005–2013	53	US-guided sclerotherapy	Ethanol 95%. Instilled and removed 75% of aspirated fluid. Re-instilled 5 to 10 mL and left in situ
Wang et al., 2015 [31]	China	Prospective cohort	2010–2012	105	US-guided sclerotherapy	Ethanol 95%. Instilled 50% of cyst fluid volume and <100 mL and left in situ.
Han et al., 2018 [32]	South Korea	Prospective cohort	2015–2017	14	Catheter-directed sclerotherapy	Ethanol 95%. Instilled 25% of cyst fluid volume and <100 mL for 20 min.
Aflatoonian et al., 2020 [33]	Iran	Retrospective cross-sectional	2013–2017	53	Sclerotherapy with retention of ethanol (n = 27); no retention (n = 16)	Ethanol 98%. Instilled 60% of cyst fluid volume for 10 min.
Miquel et al., 2020 [34]	France	Retrospective cohort	2013–2017	37	US-guided sclerotherapy	Ethanol 96%. Instilled 60% of cyst fluid volume and <60 mL for 10 min.
Huang et al., 2021 [35]	Taiwan	Retrospective cross-sectional	2008–2018	124	Sclerotherapy with retention of ethanol and CA-125 91 U/mL (n = 44); no retention and CA-125 91 U/mL (n = 80)	Ethanol 95%. Instilled variable 3–10 mL of cyst fluid volume for 1–3 min.
Lee et al., 2022 [36]	South Korea	Retrospective cohort	2014–2021	18	Catheter-directed sclerotherapy	Ethanol 99%. Instilled 25% of cyst fluid volume for 20 min.
Meng et al., 2022 [37]	China	Prospective cohort	2020–2021	70	US-guided pelvic artificial isolation with fluid	Ethanol 95%. Instilled 33% of cyst fluid volume and <60 mL for 1 min.
* Comparative studies, included for meta-analysis *						
Noma et al., 2001 [38]	Japan	Retrospective cross-sectional	1993–1998	100	US-guided sclerotherapy (n = 74); laparoscopic cystectomy (n = 26)	Ethanol. Instilled 80% of cyst fluid volume and <100 mL.
Suganuma et al., 2002 [39]	Japan	Prospective cross-sectional	N/A	59	US-guided sclerotherapy (n = 23); laparoscopic cystectomy (n = 36)	Ethanol
Yazbeck et al., 2009 [40]	France	Prospective cross-sectional	2004–2008	56	US-guided sclerotherapy (n = 35); laparoscopic cystectomy (n = 36)	Ethanol 100%. Instilled 80% of cyst fluid volume and <60 mL for 10 min.
Lee et al., 2014 [41]	South Korea	Retrospective cohort	2008–2012	65	US-guided sclerotherapy (n = 29); surgical resection (n = 36)	Ethanol 20%. Instilled and flushed 80–90% of cyst fluid volume.
Garcia-Tejedor et al., 2020 [42]	Spain	Prospective cohort pilot	2016–2018	31	US-guided aspiration plus sclerotherapy (n = 17); laparoscopic cystectomy (n = 14)	Ethanol 100%. Instilled 66% of the cyst fluid volume and <100 mL for 15 min.
Alborzi et al., 2021 [43]	Iran	Prospective cross-sectional	2013–2020	101	US-guided sclerotherapy (n = 44); laparoscopic cystectomy (n = 57)	Ethanol 96%. Instilled 80% of the cyst fluid volume and left in situ.
Koo et al., 2021 [44]	South Korea	Randomized controlled trial	2011–2019	71	Catheter-directed sclerotherapy (n = 20); surgical excision (n = 51)	Ethanol 99%. Instilled 25% of the cyst fluid volume and <100 mL for 20 min.
Martinez-Garcia et al., 2021 [45]	Spain	Prospective cohort pilot	N/A	40	US-guided aspiration plus sclerotherapy (n = 16); laparoscopic cystectomy (n = 10)	Ethanol 100%. Instilled 66% of the cyst fluid volume and <100 mL for 15 min.

† US: ultrasound; ‡ N/A: not available; § LPS: laparoscopy.

**Table 2 medicina-59-01643-t002:** Clinical outcomes after sclerotherapy of ovarian endometrioma.

* Single-Arm Studies *		
Authors, Year of Publication	Success Rate (%)	Recurrence Rate (%)
Chang et al., 1997 [17]	† N/A	46.8
Mesogitis et al., 2000 [18]	81.8	61.9
Koike et al., 2002 [19]	100	13.3
Fisch et al., 2004 [20]	75.0	0.0
Agostini et al., 2006 [21]	100	28.6
Ikuta et al., 2006 [22]	100	11.1
Hsieh et al., 2008 [23]	N/A	26.9
Gatta et al., 2010 [24]	100	8.0
André et al., 2011 [25]	N/A	N/A
Shawki et al., 2011 [26]	86.0	19.3
Wang et al., 2011 [27]	92.4	0.0
Aflatoonian et al., 2013 [28]	N/A	20.0
Garcia-Tejedor et al., 2015 [29]	100	12.1
Begum et al., 2015 [30]	79.2	11.3
Wang et al., 2015 [31]	100	0.0
Han et al., 2018 [32]	100	0.0
Aflatoonian et al., 2020 [33]	N/A	44.1
Miquel et al., 2020 [34]	87.0	2.7
Huang et al., 2021 [35]	N/A	22.5
Lee et al., 2022 [36]	100	5.5
Meng et al., 2022 [37]	100	N/A
* Comparative studies *		
Noma et al., 2001 [38]	N/A	14.9
Suganuma et al., 2002 [39]	N/A	N/A
Yazbeck et al., 2009 [40]	100	12.9
Lee et al., 2014 [41]	N/A	N/A
Garcia-Tejedor et al., 2020 [42]	N/A	5.9
Alborzi et al., 2021 [43]	N/A	34.1
Koo et al., 2021 [44]	100	0.0
Martinez-Garcia et al., 2021 [45]	N/A	N/A

† N/A: not available.

**Table 3 medicina-59-01643-t003:** Reproductive outcomes after sclerotherapy of ovarian endometrioma.

* Single-Arm Studies *	
Author, Year of Publication	Pregnancy Rate (%)
Chang et al., 1997 [17]	34.7
Mesogitis et al., 2000 [18]	† N/A
Koike et al., 2002 [19]	41.8
Fisch et al., 2004 [20]	57.0
Agostini et al., 2006 [21]	N/A
Ikuta et al., 2006 [22]	33.3
Hsieh et al., 2008 [23]	8.1
Gatta et al., 2010 [24]	6.0
André et al., 2011 [25]	20.0
Shawki et al., 2011 [26]	N/A
Wang et al., 2011 [27]	N/A
Aflatoonian et al., 2013 [28]	33.3
Garcia-Tejedor et al., 2015 [29]	0.0
Begum et al., 2015 [30]	33.9
Wang et al., 2015 [31]	N/A
Han et al., 2018 [32]	100
Aflatoonian et al., 2020 [33]	39.5
Miquel et al., 2020 [34]	37.3
Huang et al., 2021 [35]	23.3
Lee et al., 2022 [36]	N/A
Meng et al., 2022 [37]	N/A
* Comparative studies *	
Noma et al., 2001 [38]	52.2
Suganuma et al., 2002 [39]	31.4
Yazbeck et al., 2009 [40]	55.2
Lee et al., 2014 [41]	44.4
Garcia-Tejedor et al., 2020 [42]	17.6
Alborzi et al., 2021 [43]	34.1
Koo et al., 2021 [44]	N/A
Martinez-Garcia et al., 2021 [45]	42.8

† N/A: not available.

## Data Availability

Data supporting conclusions of the present study can be found in References.

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
