# Peer review of "The Efficiency of Sclerotherapy for the Management of Endometrioma: A Systematic Review and Meta-Analysis of Clinical and Fertility Outcomes"

_medicina, 2023, doi:10.3390/medicina59091643_

Round 1
Reviewer 1 Report
The article is well written with good merit.
Only few advice
Line 123
A total of 1642 patients were included in the review [17-46]—Statement is not clear, means reference number 17-46 stating a total number of 1642 patients?
The databse is 1622(Flow Chart)
Line 243
6. Patents—why this heading is there?
The Figure 2 & Figure 3 (Meta analysis) outcome advised to include in the discussion and conclusion section.
The significance of Funnel plots in appendix need to describe in discussion section.
Advised to add Future Direction or Scope
Author Response
Thank You for taking the time to review our manuscript, and for Your comments. They are
crucial and valuable to us in raising the quality standard of our work. In addition, a specification for
Your revisions is below:
“Line 123
A total of 1642 patients were included in the review [17-46]—Statement is not clear, means
reference number 17-46 stating a total number of 1642 patients?
The database is 1622(Flow Chart)”
- Thank You for Your comment. The patients included in our study are 1642. Table 1 shows
the number of patients enrolled in each trial considered for our systematic review, and the
total is 1642.
“Line 243
6. Patents—why this heading is there?”
- Thank You for Your observation. We changed the main text in its respect.
“The Figure 2 & Figure 3 (Meta analysis) outcome advised to include in the discussion and
conclusion section.”
- Thank You for Your brilliant observation. Although, we discussed the outcomes from Figure
2 and Figure 3 in lines: 187-to-193, 199 and 200 (Figure 2); and 211-to-214 (Figure 3). We
enriched the Conclusions in lines 257-259.
“The significance of Funnel plots in appendix need to describe in discussion section.”
- Thank You for Your advice. We discussed the significance of Funnel Plots in the Discussion
and Conclusions sections. Please see lines 204 and 205, 213-215, 259.
“Advised to add Future Direction or Scope”
- Thank You for Your advice. We discussed the options of liquid biopsy and microbial analysis
in identifying the best candidates for sclerotherapy in lines 248-to-254.
Also, You can find the rewritten and corrected manuscript version in the attached file. We
highlighted any changes made. Thank You very much for Your advice and comments. We hope we
have complied with Your requests.
Reviewer 2 Report
The introduction section is so simple, that it needs more to explain.
Abbreviations should be defined at first mention within the whole text.
Figure 1. Flow Diagram………..The last 2 boxes are the same “studies included in quantitative synthesis” Is there a difference? Please check and correct…..They have other descriptions within the text.
References within tables must be cited in number [ ] according to journal guidelines.
Figure 2. consists of both table and figure……..I suggest to separate both from each other.
Figure 3. consists of both table and figure……..I suggest to separate both from each other.
Page 11, Line 170: The reported data show that the is a high SR technique…..correct sentence
Page 12, Line 197: revealed by modifications of AMH…………..define abbreviation in the first time.
Abstract, Line 14: Sclerotherapy consists in injecting alcohol into the endometrioma to destroy the pseudocapsule…….correct sentence.
The abstract needs to be more concise with the main significant findings.
Good work and best regards
Good
Author Response
Dear Reviewer (#2),
Thank You for taking the time to review our manuscript, and for Your comments. They are
crucial and valuable to us in raising the quality standard of our work. In addition, a specification for
Your revisions is below:
“The introduction section is so simple, that it needs more to explain.”
- Thank You for Your comments. We enriched the Introduction section with further
information about the use of sclerotherapy.
“Abbreviations should be defined at first mention within the whole text.”
- Thank You for Your observation. We defined abbreviations mentioned in the main text.
“Figure 1. Flow Diagram………..The last 2 boxes are the same “studies included in quantitative
synthesis” Is there a difference? Please check and correct…..They have other descriptions within the
text.”
- Thank You for Your brilliant observation. We corrected the definition in the box, to
highlight the difference between the quantitative and qualitative syntheses.
“References within tables must be cited in number [ ] according to journal guidelines.”
- Thank You for You comment. We cited references in brackets in Table 1, Table 2, Table 3,
and Appendix A.
“Figure 2. consists of both table and figure……..I suggest to separate both from each other.
Figure 3. consists of both table and figure……..I suggest to separate both from each other.”
- Thank You for Your advice. Although, we chose to provide aligned tables and figures for
each outcome, in order to have a complete representation of: events number, Confidence
Intervals, and Odds Ratios.
“Page 11, Line 170: The reported data show that the is a high SR technique…..correct sentence”
- Thank You for Your advice. We corrected the sentence.
“Abstract, Line 14: Sclerotherapy consists in injecting alcohol into the endometrioma to destroy the
pseudocapsule…….correct sentence.”
- Thank You for Your advice. We modified the sentence. Please, see line 15.
“Page 12, Line 197: revealed by modifications of AMH…………..define abbreviation in the first
time.”
- Thank You for Your observation. We defined the abbreviation “AMH” – mentioned in the
main text – as Anti-Müllerian Hormone.
“The abstract needs to be more concise with the main significant findings.”
- Thank You for Your comment. We changed the abstract section according to Your
observation, and added the most significant findings.
Also, you can find the rewritten and corrected manuscript version in the attached file. We
highlighted any changes made. Thank You very much for Your advice and comments. We hope we
have complied with Your requests.